# Meningeal lymphoid structures are activated under acute and chronic spinal cord pathologies

Merav Cohen[1],*, Amir Giladi[1],*, Catarina Raposo[2],*, Mor Zada[1], Baoguo Li[1], Julia Ruckh[2], Aleksandra Deczkowska[1], Boaz Mohar[2], Ravid Shechter[2], Rachel G Lichtenstein[3], Ido Amit[1], Michal Schwartz[2,4]

Tertiary lymphoid structures (TLS) are organized aggregates of B and T cells formed ectopically during different stages of life in response to inflammation, infection, or cancer. Here, we describe formation of structures reminiscent of TLS in the spinal cord meninges under several central nervous system (CNS) pathologies. After acute spinal cord injury, B and T lymphocytes locally aggregate within the meninges to form TLS-like structures, and continue to accumulate during the late phase of the response to the injury, with a negative impact on subsequent pathological conditions, such as experimental autoimmune encephalomyelitis. Using a chronic model of spinal cord pathology, the mSOD1 mouse model of amyotrophic lateral sclerosis, we further showed by single-cell RNA-sequencing that a meningeal lymphocyte niche forms, with a unique organization and activation state, including accumulation of pre-B cells in the spinal cord meninges. Such a response was not found in the CNS-draining cervical lymph nodes. The present findings suggest that a special immune response develops in the meninges during various neurological pathologies in the CNS, a possible reflection of its immune privileged nature.

## Introduction

Following any deviation from homeostasis in the central nervous system (CNS), the immune response is activated to facilitate repair and to resolve the detrimental parenchymal inflammation. Despite the privileged nature of the CNS, many immunological process take place within its boundaries, both in homeostasis and under pathological conditions, with similarities to those that occur in the periphery (Lalancette-Hebert et al, 2007; Liesz et al, 2009; Shechter et al, 2009; David & Kroner, 2011; London et al, 2011, 2013; Martino et al, 2011; Michaud et al, 2013; Cohen et al, 2014, 2017; Peruzzotti-Jametti et al, 2014; Raposo et al, 2014; Kunis et al, 2015; Russo & McGavern, 2015).

Tertiary lymphoid structures (TLS) are ectopic lymphoid aggregates formed locally in non-lymphoid tissues after organ development, induced by pathologies characterized by ongoing chronic inflammation, infection, autoimmunity and cancer (Kendall et al, 2007; GeurtsvanKessel et al, 2009; de Chaisemartin et al, 2011; Dieu-Nosjean et al, 2014). TLS share cellular and organizational features with secondary lymphoid organs (SLOs), including segregation of B- and T-cell areas that are supported by stromal/vascular components, and presence of activated germinal centers (GCs) with follicular DCs, where clonal expansion, somatic mutation, and isotype switching can occur (Gommerman & Browning, 2003). Their formation has been proposed to involve defective lymphatic drainage and continuous local antigen stimulation (Thaunat et al, 2006).

Recent studies in the field of CNS lymphatic drainage revealed that in homeostasis, a lymphatic vasculature is present in the dural sinuses of the brain meninges (Weller et al, 2009; Aspelund et al, 2015; Louveau et al, 2015), in addition to the perivascular pathways that drain interstitial fluid from the brain parenchyma, and cerebrospinal fluid (CSF) from the subarachnoid space to cervical lymph nodes (cLNs) (Carare et al, 2014; Lochhead et al, 2015). These observations, together with our findings that entry of immuno-regulatory cells to sites of injured CNS parenchyma is orchestrated through the brain's border, the choroid plexus (CP) epithelium (Kunis et al, 2013; Shechter et al, 2013; Raposo et al, 2014), suggest that the meningeal lymphatic niche might become activated and participate in the immunological network triggered by CNS injury. TLS are widespread across different tissues and under different pathological conditions in the periphery. Within the CNS, their formation in the form of B-cell aggregates that execute a GC response was mainly identified in brain meninges under inflammatory diseases of the brain, such as multiple sclerosis, and its mouse model, experimental autoimmune encephalomyelitis (EAE) (Columba-Cabezas et al, 2006; Howell et al, 2011; Peters et al, 2011; Kuerten et al, 2012; Pikor et al, 2015; Lehmann-Horn et al, 2016; Bevan et al, 2018; Magliozzi et al, 2019).

[1]Department of Immunology, Weizmann Institute of Science, Rehovot, Israel  [2]Department of Neurobiology, Weizmann Institute of Science, Rehovot, Israel  [3]Avram and Stella Goldstein-Goren Department of Biotechnology Engineering, Ben-Gurion University of the Negev, Beersheba, Israel  [4]Klarman Cell Observatory, Broad Institute of Massachusetts Institute of Technology (MIT) and Harvard, Cambridge, MA, USA

Correspondence: michal.schwartz@weizmann.ac.il; merav.cohen@weizmann.ac.il
*Merav Cohen, Amir Giladi, and Catarina Raposo contributed equally to this work

Here, we examined the response of the spinal cord meningeal lymphocyte compartment to acute injury (spinal cord injury; SCI), as well as to chronic neurodegenerative conditions. We found that acute CNS injury is followed by formation of structures, reminiscent of TLS, within the spine meninges in close proximity to the lesion site. Such meningeal structures were found to be formed during the chronic phase of the response to acute injury, and mainly in the pia and dura mater. Meningeal TLS-like lymphocytes acquire an inflammatory phenotype, different from that observed in peripheral draining cLN. By using the mSOD1 mouse model of chronic spinal cord degeneration, often used as a model of amyotrophic lateral sclerosis (ALS), we found that lymphocytes isolated from spine meninges, but not from draining cLN, showed dynamic changes in lymphocyte activation, and specifically in B-cell differentiation (pre-B cells and plasma cells), along disease progression.

# Results

### Immunological niche in the meninges after acute SCI

SCI induces an acute immune response in the injured parenchyma (Rolls et al, 2008; Shechter et al, 2009). When exploring the immune response within the lesion site 14 d after SCI (Raposo et al, 2014), we found aggregates of TCRβ⁺ T cells in the spinal cord meninges, in close proximity to the primary injury site (Fig 1A). To fully understand the spatial organization of immune cells in the spinal cord meninges, we performed scanning electron microscopy (SEM) of the spinal cord parenchyma, together with the bones that envelop it. This technique enabled us to examine the meningeal tissue located between the spinal cord parenchyma and the bone, and to identify cells associated with these meningeal layers (Fig 1B). SEM images showed differences in cellular and matrix composition between meninges derived from uninjured spinal cords, compared with meninges isolated 14 d after SCI. Scanning of the meninges at high magnification indicated that SCI induced the appearance of areas with fiber-like structures, compared to the smooth meningeal tissue in the uninjured animals (Fig 1C). Importantly, this structural feature of fiber-like meningeal components appeared together with leukocyte accumulation, specifically within meninges isolated from spinally injured mice (Fig 1C).

For quantitative characterization of the lymphocyte populations in the spinal cord meninges after SCI, we first calibrated a tissue dissociation and cell purification technique for this tissue (see the Materials and Methods section). We then compared CD3⁺ T- and B220⁺ B-cell frequencies between the meninges, the spinal cord parenchyma, and the peripheral cervical lymph nodes (cLN) (Fig 1D). We found that the lymphocyte composition in the spinal cord meninges was distinct from that observed within the injured parenchyma, and more closely related to cLN (Fig 1E). Whereas the lesion site of injured animals contained a majority of T cells (89%, predominantly CD4⁺ T cells), in the adjacent meninges, as well as in the draining cLN, we found equal proportions of B versus T cells, and a similar ratio of CD4⁺/CD4⁻ T cells (Fig 1D and E). Next, we quantified the numbers of CD45⁺ leukocytes, CD3⁺ T cells, B220⁺ B cells, and CD4⁺ T cells within a 6-mm length of meningeal tissue, in close proximity to the lesion site, compared with distal regions in the spinal cord parenchyma, at day 14 after SCI (Fig 1F). We found that SCI induced the accumulation of CD45⁺ cells in

general, and specifically of B and T cells, mainly of the CD4⁺ subset, in a spatially restricted manner; the increase in lymphocyte numbers was observed in meningeal tissues that were isolated from areas in close proximity to the lesion site, and not from distal regions (Fig 1G). Taken together, these results suggest that during the late phase of the response to SCI, a lymphocyte-driven immune response develops in the adjacent meninges, characterized by a cLN-like immune cell composition.

### The meningeal immune response within its three anatomical layers

The meninges are composed of three layers, the dura, arachnoid, and pia mater. Lymphocyte accumulation in the spinal cord meninges at the lesion site area after SCI raised the question of whether the meningeal immune response is uniform in all three layers, or whether any of the meningeal layers hosts a unique immunological niche. We performed a detailed analysis of the meningeal immune cell composition by microdissection of the dura, arachnoid, and pia spinal cord meningeal layers (Figs 2A and S1), and quantified the number of cell clusters (aggregates with more than 10 cells) over time after SCI. Because the pia and dura layers were enriched with lymphocyte clusters, compared with arachnoid layer, during the chronic phase of the response to injury, we chose to further focus on these two layers.

To gain insight regarding the spatial aggregations and distribution of lymphocytes within the meningeal layers, we performed immunostaining of whole-mount dissected pia and dura mater. Under uninjured conditions, CD3⁺ T and B220⁺ B cells were abundant in the pia and dura mater, respectively, and were randomly distributed in the tissue, without any organized structures (Fig 2B). At day 14 after SCI, in the meningeal tissue located in close proximity to the lesion site, we were able to detect the accumulation of lymphocyte aggregates and formation of large well-organized clusters in the pia and the dura mater (Fig 2C and D). We next quantified the lymphocyte composition in the clusters that were observed in the pia and dura mater. We found that in uninjured animals and at day 7 after SCI, T cells were predominant in the pia, whereas B cells were enriched in the dura (Fig 2E). From day 14 onward, the late and chronic phase of response to SCI, each layer contained clusters composed of heterogeneous lymphocyte populations (Fig 2E). Thus, the immune response in the meninges is spatially regulated and continues to evolve during the chronic phase of the response to acute insult. Overall, it appears that acute CNS insult induces deviation in lymphocyte composition within the different meningeal layers. Of note, we cannot rule out the possibility of some cross-contamination between layers during microdissection.

### Hallmarks of lymphogenesis factors present in the meninges after SCI

To understand the signaling that could potentially lead to lymphocyte accumulation at the meninges, we measured expression of factors associated with recruitment, accumulation, and induction of lymphocyte organization in non-lymphoid organs, such as the chemokines CXCL13,

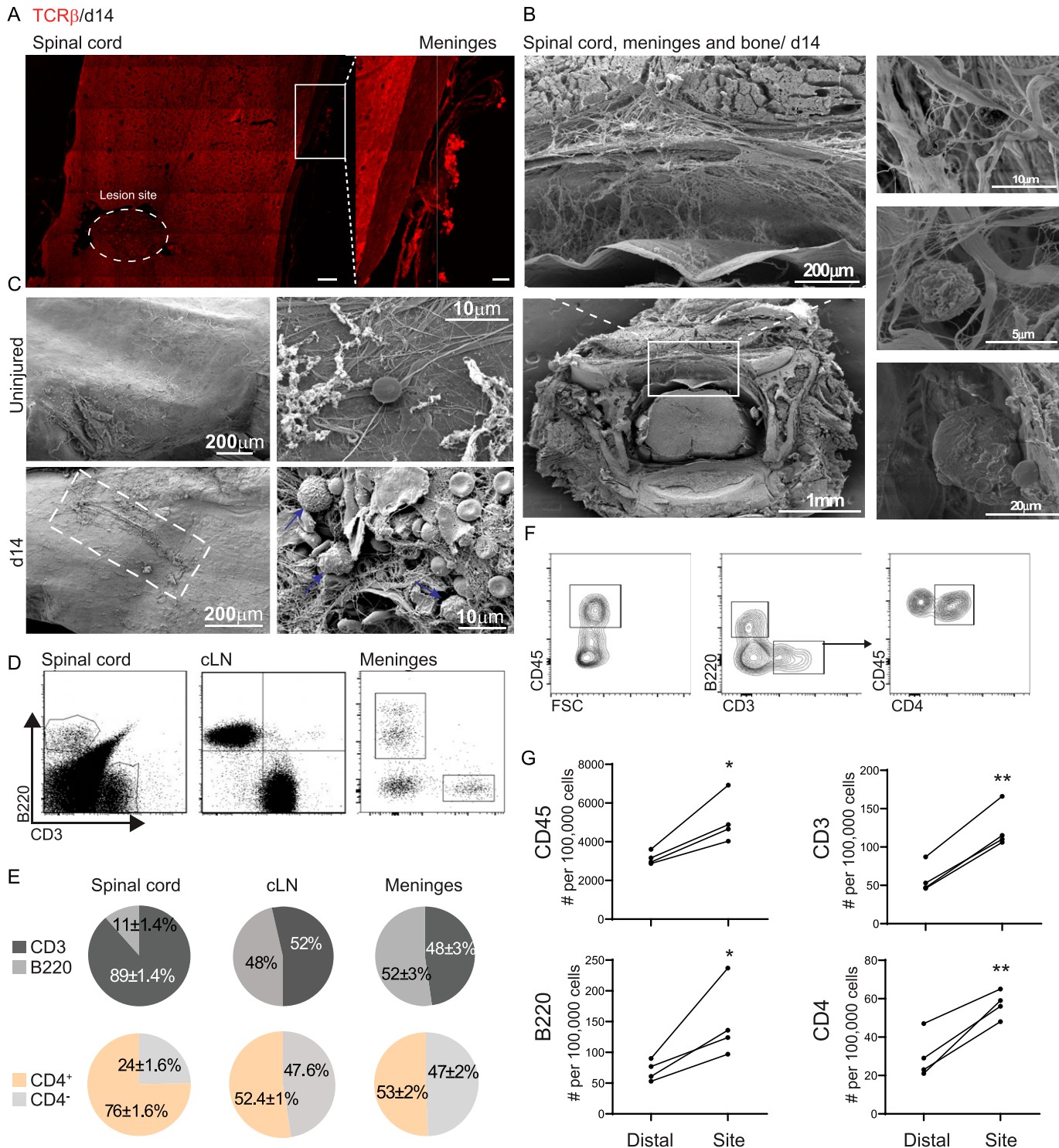

**Figure 1. Lymphocyte aggregation in the spinal cord meninges after spinal cord injury (SCI).**
**(A)** Representative immunofluorescent staining of TCRβ⁺ cells in the spinal cord parenchyma and the adjacent meninges 14 d following SCI; scale bar: 100, 20 μm inset. **(B)** Transverse scanning electron microscopy image of spinal cord including the bone 14 d after SCI, showing separation of meningeal layers and the presence of cells on the meninges. **(C)** Scanning electron microscopy images of spinal cord meninges derived from uninjured animals (upper panel) and at day 14 after SCI (lower panel), taken at different magnifications. **(D)** Representative flow cytometry plots demonstrating the composition of B and T cells at the spinal cord parenchyma, draining cervical lymph node, and spinal cord meninges. n = 4; each sample represents a pool of two animals. **(E)** Flow cytometry quantification of T and B cell ratio out of total lymphocytes (upper panel), and of CD4⁺ and CD4⁻ out of total T cells (lower panel) in the spinal cord parenchyma, cervical lymph node and spinal cord meninges at day 14 following SCI. n = 4; each sample represents a pool of two animals. Data shown are representative of at least three independent experiments. **(F)** Representative flow cytometry plots demonstrating the gating strategy for characterization of the leukocyte and lymphocyte populations in the spinal cord meninges. n = 4; each sample represents a pool of

CCL21, CCL19, and proteins related to the lymphotoxin (LT) pathway (Kratz et al, 1996; Fan et al, 2000; Amft et al, 2001; Shi et al, 2001). Using quantitative RT-PCR (qRT-PCR) analysis of the pooled meningeal tissue at different time points after SCI, we found increased expression of several of these molecules, including *Ccl21*, lymphotoxin-α (*Ltα*), and lymphotoxin-β (*Ltβ*), and their corresponding receptors (*Ccr7* and *Ltβr*), at the late phase of the response to injury (days 14 and 19), relative to the meninges of uninjured animals (Fig 3A). In addition, using whole-mount immunostaining, we found that some of these molecules, such as CCL21, were associated with CD31⁺ endothelium in the brain meninges during homeostasis, and also following SCI (Fig 3B). In the spinal cord meninges, we identified the expression of CCL21 (Fig 3C), and of the lymphatic vessel marker, LYVE-1, next to CD31⁺CD34⁺ endothelial cells in the pia (Fig 3D) and the dura (Fig 3E) mater 14 d after SCI. These results provide an evidence of an infrastructure which can support lymphocyte homing to the meningeal layers following CNS insult (Baluk & McDonald, 2008; Mueller & Germain, 2009). In addition, in the dura mater, we also found areas of high immunoreactivity of the B cell chemokine CXCL13, associated with clusters of B cells (Fig 3F). Notably, the presence of the chemokine CXCL13 in the dura mater is in line with the observations that the dura compartment is more closely associated with a B-cell response (Fig 2B and E). We quantified the number of B-cell clusters co-localized with CXCL13 at the acute (7 d after SCI) and chronic phases (day 14 and 21 after SCI), and found an increase in these clusters during the late stages of the response to injury (Fig 3G). Notably, CXCL13, like CCL21, was also detected in the dura mater of uninjured mice (Fig 3A), but without co-localization with B-cell aggregates, suggesting that in homeostasis, the meninges are endowed with an infrastructure that enables signaling for lymphoid niche formation upon need; however, formation of lymphocyte aggregates is a consequence of multiple signals.

**Meningeal lymphocyte structures are reminiscent of SLOs**

Our results showed that the immune response occurring in the spinal cord meninges is initiated by the homing of lymphocytes to an area located in close proximity to the lesion site. To further characterize the immune fate of these meningeal lymphocyte clusters, we sorted B220⁺ B and CD4⁺ T cells from the meninges adjacent to the lesion site and compared their cytokine and transcription factor gene expression profiles to those of lymphocytes derived from the CNS-draining cLN, 14 d after SCI. We found that meningeal B cells expressed genes related to plasma cell development and differentiation, such as *Pax5*, *Irf4*, and *Xbp1* (Nutt et al, 2015), similar to B cells isolated from cLN (Fig 4A). Interestingly, meningeal B cells exhibited a strong plasma cell response (Xbp1^high), and a pro-inflammatory activation state characterized by expression of genes related to CNS inflammation and autoimmunity, including *Il6*, *Ifnγ*, *Ifnγr*, *Il23*, and *Tnfα* (Harris et al, 2000, 2005; Duddy et al, 2007; Barr et al, 2012). This plasma cell response and pro-

inflammatory phenotype were attenuated and absent, respectively, in B cells isolated from cLN (Fig 4A). Meningeal CD4⁺ T cells showed high heterogeneity in cellular phenotype, exhibiting increased gene expression of regulatory (*Foxp3* and *Il10*), Th1 (*Tbx21* and *Ifnγ*), and encephalitogenic (*Rorc*, *Il6*, and *Tnfα*) (Korn et al, 2008) markers. As was shown for the B-cell compartment, the expression of CD4⁺ T-cell immune activation genes was restricted to the spine meninges, and absent from the peripheral cLN (Fig 4A).

GC is a defining marker of B cell clusters, and a fundamental characteristic of any SLO. GC can be formed ectopically and are found under conditions of chronic inflammation as part of a local lymphoid organization, termed TLS (Schroder et al, 1996; Luther et al, 2000; Shi et al, 2001; Moyron-Quiroz et al, 2004; Serafini et al, 2004; Grabner et al, 2009; Shields et al, 2010; von Budingen et al, 2015). We, therefore, evaluated the maturation state of the meningeal B-cell population after SCI, to detect the possible presence of GC. We found that meningeal B cells derived from both uninjured and injured animals consisted of both B220^high and B220^low populations (Fig 4B). Interestingly, the non-conventional B220^low subset, which was shown to mainly include cells mediating an antibody response (Kawabe et al, 2004; Tachikawa et al, 2008; Fujii et al, 2010), was the most abundant in the lesion site meninges and showed the strongest induction after SCI (Fig 4B). Using immunostaining, we identified IgD⁺ and IgM⁺ B cells in the meninges of injured mice (Fig 4C and D). Moreover, flow cytometry analysis of lesion site meninges at day 14 after SCI revealed the presence of B cells at different stages of isotype switching. The mature but naïve state, characterized by IgM⁺IgD⁺ B cells (Dalakas, 2008), was derived from the B220^high B cells, whereas B220^low cells were all IgD⁻, including both IgM⁺ and IgM⁻ cells, reflecting mature and more activated states (Fig 4E). In addition, we found that as a result of SCI, the meningeal B cells, mainly the B220^low population, expressed the GC activation markers peanut agglutinin (PNA) and GL7 (Fig 4F).

An important feature of an active lymphoid site is expansion of lymphocyte clones, in association with antigen presentation. We identified proliferating Ki67⁺ B cells, in close proximity to non-proliferating ones within the same cluster, 14 d after SCI (Fig 4G). Moreover, bromodeoxyuridine (BrdU) injection to spinally injured mice, 12 and 1 h before their meninges were excised, showed that most CD3⁺ T cells at the meninges were in a proliferative state, incorporating BrdU, with only a few non-proliferating cells (Fig 4H). Notably, we found that the meningeal lymphocyte clusters were also populated by CD11c⁺ cells next to T cell areas, indicating the occurrence of possible immunological synapses between DC and T cells within the meningeal lymphocyte structures (Fig 4I); such interactions could drive the proliferation of the T cells, an important feature of LN immune surveillance.

We further analyzed the phenotype of the meningeal T cells and compared it with that of T cells in the CNS-draining cLN. Analysis of the meningeal CD4⁺ T cells at day 14 after SCI revealed that CD44^high memory T cells predominated in the meningeal lymphocyte structures (87% of CD4⁺TCRβ⁺ cells), while this memory population was diminished in cLN (21% of CD4⁺TCRβ⁺ cells; Fig 4J). Importantly, the majority of memory T cells in the meningeal lymphoid structures lost

two animals. **(G)** Flow cytometry quantification of leukocytes and lymphocyte populations present at the lesion site meninges versus those located distally to the lesion site; each sample represents pooled tissue from two animals. *P < 0.05. **P < 0.01. Data are presented as mean ± SEM.

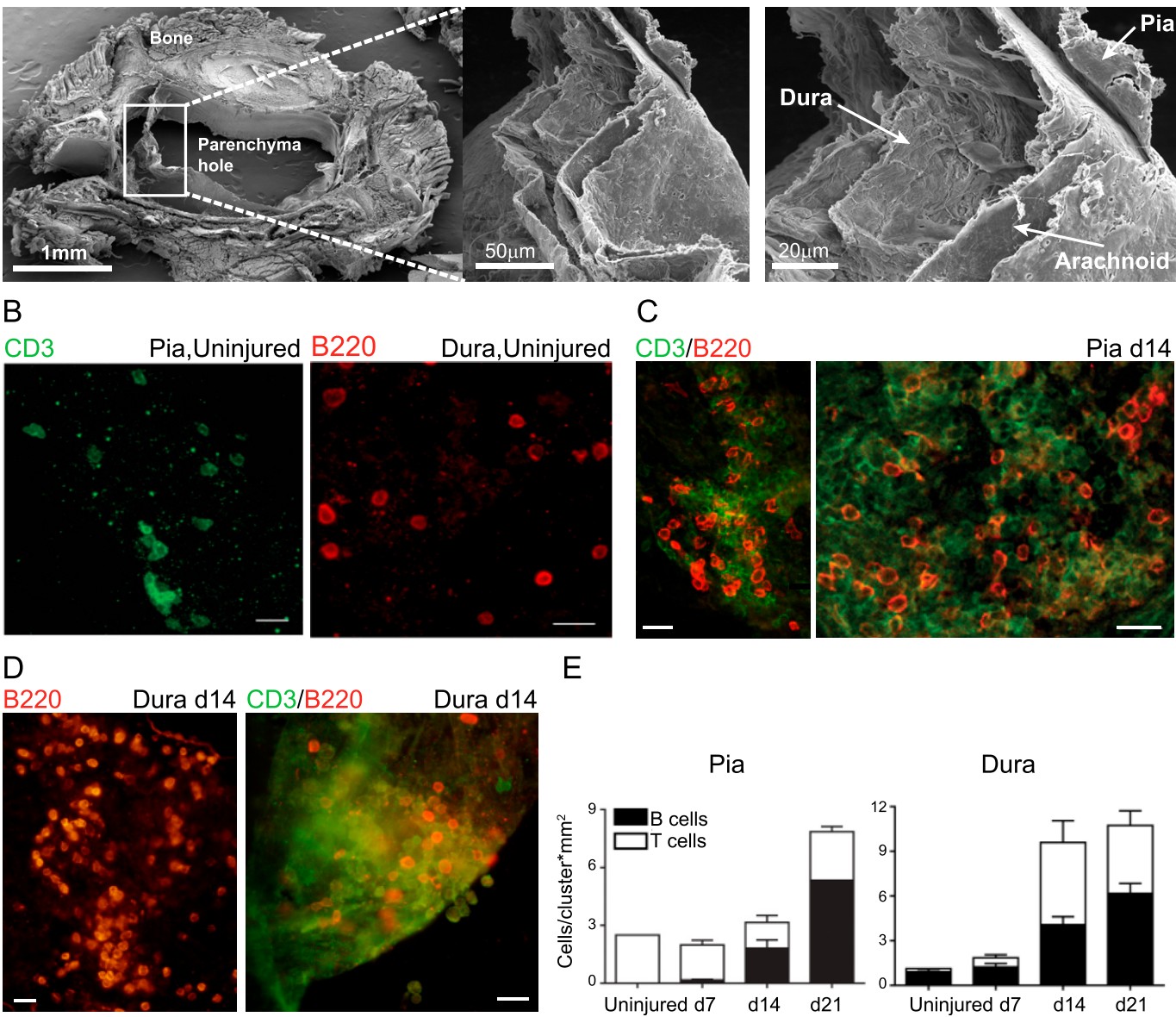

**Figure 2. Segregation of the meningeal lymphocyte niche into the anatomical layers.**
**(A)** Representative scanning electron microscopy (SEM) images showing separation of the three spinal cord meningeal layers. **(B)** Representative immunofluorescent whole-mount staining of the spinal cord pia and dura meningeal layers, for T cells and B cells under uninjured conditions. Scale bar: 20 $\mu$m. **(C, D)** Representative immunofluorescent whole-mount staining of the pia (C), and dura (D) meningeal B- and T-cell clusters at d 14 after SCI. Scale bar; 50 $\mu$m. **(E)** Whole-mount quantification of the number of B cells (black) or T cells (white) in each cluster present at the pia and dura layers of uninjured mice, and along different time points following SCI. **(E)** Results are pooled from three animals at each time point, and normalized to the indicated area.

the expression of L-selectin, and therefore exhibited a phenotype of effector-like memory T cells (76%; CD44$^{high}$CD62L$^{low}$), whereas only 11% were central-like memory T cells (CD44$^{high}$CD62L$^{high}$). Interestingly, in cLN, CD44$^+$ T cells comprised equal proportions of T effector-like and T central-like memory cells (Fig 4J). Overall, T-effector memory cells represented 76% of the total meningeal CD4$^+$ T cells, compared with 40% found in the adjacent injured parenchyma (Raposo et al, 2014). Thus, we observed that spinal cord meninges harbor a unique lymphoid niche after injury, at a specific spatial location relative to the lesion site. The particular features of the meningeal lymphocyte clusters, including their gene expression profile, presence of B cells with GC features, proliferation of T lymphocytes and their co-localization with CD11c$^+$ DC, enrichment of effector-like memory T cells, and the fact that these clusters showed a stronger immune activation state than the peripheral cLN following SCI, led us to define them as meningeal TLS.

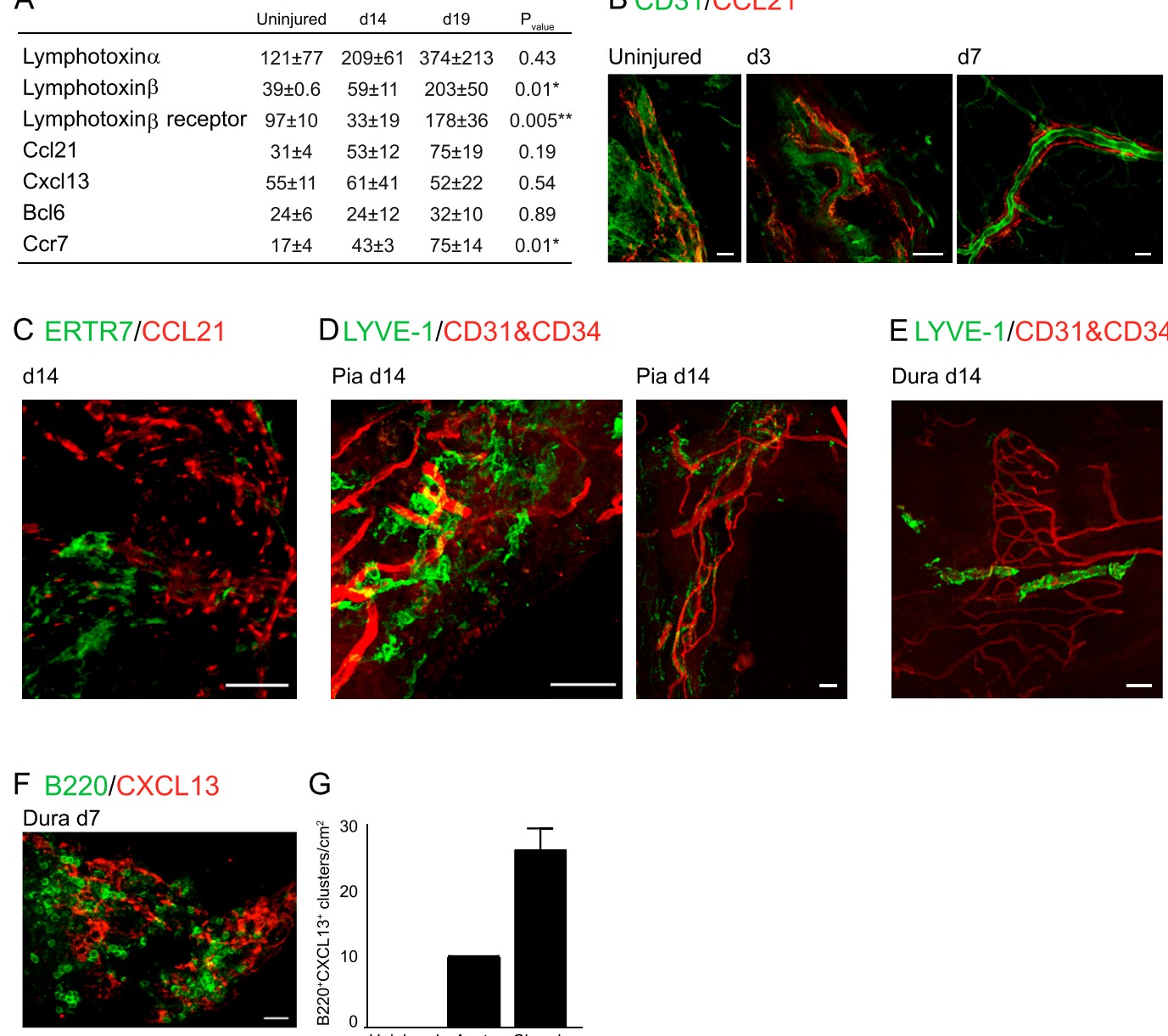

## A

| | Uninjured | d14 | d19 | P$_{value}$ |
|---|---|---|---|---|
| Lymphotoxinα | 121±77 | 209±61 | 374±213 | 0.43 |
| Lymphotoxinβ | 39±0.6 | 59±11 | 203±50 | 0.01* |
| Lymphotoxinβ receptor | 97±10 | 33±19 | 178±36 | 0.005** |
| Ccl21 | 31±4 | 53±12 | 75±19 | 0.19 |
| Cxcl13 | 55±11 | 61±41 | 52±22 | 0.54 |
| Bcl6 | 24±6 | 24±12 | 32±10 | 0.89 |
| Ccr7 | 17±4 | 43±3 | 75±14 | 0.01* |

## B CD31/CCL21

Uninjured   d3   d7

## C ERTR7/CCL21

d14

## D LYVE-1/CD31&CD34

Pia d14   Pia d14

## E LYVE-1/CD31&CD34

Dura d14

## F B220/CXCL13

Dura d7

## G

**Figure 3.  Presence of lymphogenesis factors in the spinal cord meninges.**
**(A)** Gene expression evaluation of lymphogenesis factors in spinal cord meninges adjacent to the injury site, at different time points after the insult, and in uninjured mice, n = 2–4; each sample represents a pool of three animals. ANOVA$_{Lta}$; F = 1.1, P = 0.43. ANOVA$_{Ltβ}$; F = 12.9, P = 0.01. ANOVA$_{Ccl21}$; F = 2.3, P = 0.18. ANOVA$_{Cxcl13}$; F = 0.7, P = 0.54. ANOVA$_{Bcl6}$; F = 0.11, P = 0.89. ANOVA$_{Ccr7}$; F = 10.3, P = 0.01. ANOVA$_{Ltβr}$; F = 14.5, P = 0.005. **(B)** Whole-mount staining of CD31 and CCL21 in brain meninges, at days 3 and 7 after SCI, and in uninjured mice. Uninjured and day 3 sections: scale bar; 50 μm; day 7 section: scale bar; 100 μm. **(C)** Immunofluorescent whole-mount staining of spinal cord meninges 14 d after SCI for reticular fibroblasts (ERTR7) and the T-cell chemoattractant molecule, CCL21. Scale bar: 50 μm. **(D, E)** Immunofluorescent whole-mount staining of spinal cord meninges 14 d after SCI for lymphatic vessels (LYVE-1) and endothelium (CD31, CD34) in the pia (D), and the dura (E). Scale bar; 50 μm. **(F, G)** Immunofluorescent whole-mount staining of dura mater at d 7 after SCI for B220+ B cells, and for the B cell-associated chemokine, CXCL13, and (G) quantification of the number of co-localized B cell clusters with CXCL13 expression, under uninjured conditions, and at acute (day 7 after SCI) and chronic (days 14 and 21 after SCI) phases of the response to the injury. Results are pooled from three animals at each time point, and normalized to the tissue area. Scale bar; 50 μm. Asterisk in (A) indicates statistically significant differences after ANOVA post-hoc analysis using Tukey's honestly significant difference (*, relative to uninjured; ** in *LTβr*, relative to d 14 after SCI) *P < 0.05. **P < 0.01. Data are presented as mean ± SEM.

## Impact of meningeal TLS on CNS pathologies

Based on our results, we hypothesized that the meningeal TLS might play a role in chronic CNS pathologies in which inflammation is part of the disease etiology. To test this possibility, we examined the meningeal immune response under the inflammatory condition, EAE. We immunized mice with the encephalitogenic peptide (35–55) derived from myelin oligodendrocyte glycoprotein (MOG), and analyzed the inflammatory and encephalitogenic state of the meningeal lymphocytes under three different conditions: 14 d after SCI (SCI), 14 d after EAE induction (EAE), and 14

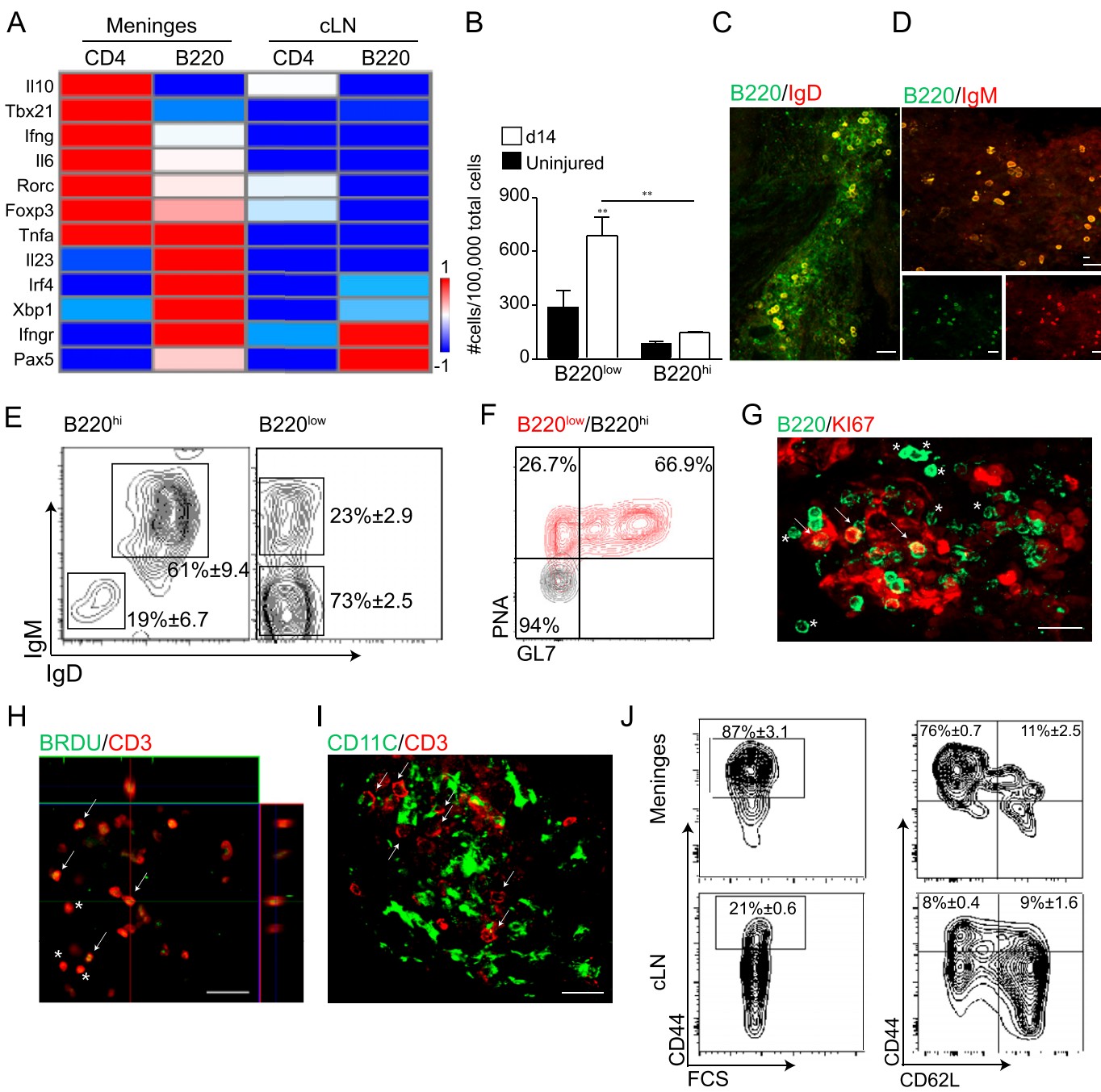

**Figure 4. Meningeal lymphocytes are activated and share features of tertiary lymphoid structures.**
**(A)** Gene expression analysis of isolated B220$^+$ B and CD4$^+$ T cells, from the spinal cord meninges and cervical lymph node 14 d after SCI, for evaluation of cytokines and transcription factors associated with inflammation and lymphocyte activation. Tissues from 20 animals were pooled; data shown are representative of two independent experiments; red and blue indicate high and low relative expression, respectively. **(B)** Quantification of B220$^{low}$ and B220$^{high}$ cells at the meninges in uninjured animals (black) and at day 14 after SCI (white); n = 2–4; each sample represents a pool of four animals. Data shown are representative of four independent experiments. **(C, D)** Immunofluorescent whole-mount staining of spinal cord meninges for IgD$^+$ (C), and IgM$^+$ (D) B cells at d 14 after SCI. Scale bar; 50 $\mu$m. **(E)** Flow cytometry plots indicating composition of IgM$^+$ and IgD$^+$ cells out of B220$^{low}$ and B220$^{high}$ B-cell subsets at the spinal cord meninges, n = 2 samples; each sample represents a pool of two animals. Data shown are representative of two independent experiments. **(F)** Density flow cytometry plot demonstrating the presence of PNA$^+$GL7$^+$ GC B cells at the meninges at d 14 after SCI. Meningeal tissues were pooled from four animals. **(G)** Representative immunofluorescent whole-mount staining at d 14 after SCI, of dura mater samples for proliferating Ki67$^+$ cells and B cells. Asterisks indicate non-proliferating B cells. Scale bar; 50 $\mu$m. **(H)** Representative immunofluorescent whole-mount staining of meninges, demonstrating the proliferating BrdU$^+$CD3$^+$ T cells. Asterisks indicate non-proliferating T cells. Scale bar; 50 $\mu$m. **(I)** Representative immunofluorescent whole-mount staining of meningeal CD11c$^+$ DC in close proximity to CD3$^+$ T cells at d 14 after SCI. Scale bar; 50 $\mu$m. **(J)** Flow cytometry density plots of CD4$^+$ T cells isolated from cLN and meninges adjacent to the lesion site 14 d after SCI, comparing the composition of CD44$^{high}$ memory T cells, and the subpopulations of central-like and effector-like memory T cells. Results are representative of two experiments, n = 2; each sample represents a pool of four animals. **P < 0.01; Data are presented as mean ± SEM.

d after EAE induced 14 d after SCI (SCI+EAE) (Fig 5A). Induction of EAE, a condition in which the T-cell parenchymal response has been characterized in detail, induced an enhanced accumulation of T lymphocytes in the spinal cord meninges, compared with meninges of mice that were subjected to SCI alone, whereas the meningeal B-cell niche was unchanged (Fig 5B). Importantly, meninges of mice that were spinally injured and then injected with MOG peptide for EAE induction (SCI+EAE), at a time when there was a well-formed TLS following the injury, exhibited increased accumulation of both B and T lymphocytes (Fig 5B). Gene expression analysis of T and B lymphocytes isolated from the spinal cord meninges under the three conditions revealed a general trend of higher encephalitogenic Th17 response (e.g., *Rorc*, *Il17a*, *Irf4*, *Pdpn*, *Il17ra*, and *Il6*) (Brustle et al, 2007) in the meninges of mice subjected to SCI+EAE, relative to lymphocytes isolated from meninges of mice subjected to SCI or EAE alone (Fig 5C and D).

## Meningeal lymphocytes are activated in a model of ALS

Based on the above results and to examine to what extent the observed phenomenon of meningeal TLS is also an outcome of chronic neurodegenerative spinal cord pathologies that are not

inflammatory in etiology, we used an animal model of ALS. Specifically, we used a transgenic mouse model for ALS, bearing a mutation in the ubiquitously expressed Cu-Zn superoxide dismutase (SOD1), and isolated all T and B lymphocytes from their spinal cord meninges at four stages of disease progression (92, 106, 136, and 161 d). Disease progression was associated with a decrease in body weight of mSOD1 mice (Fig S2A).

To thoroughly characterize the gene expression profile of different lymphocyte subsets derived from the spinal cord meninges, relative to CNS-draining cLN along disease progression, we recalibrated the protocol for dissociation of the meningeal tissue (see the Materials and Methods section), and sorted B and T lymphocytes (Fig S2B) to perform massively parallel single-cell RNA-sequencing (MARS-seq) (Keren-Shaul et al, 2019). We profiled 1051 B220⁺ B cells and 946 CD3⁺ T cells from the meninges and cLN of 16 SOD1 mice and used the MetaCell algorithm to identify homogeneous and robust groups of cells ("metacells") (Baran et al, 2019). A heat-map representation of single-cell gene expression profiles revealed a clear separation between B and T lineages, and a subdivision of the cells into diverse cell subsets (Fig 6A). In the T-cell compartment, we identified Ccl5⁺ T cells (characterized by

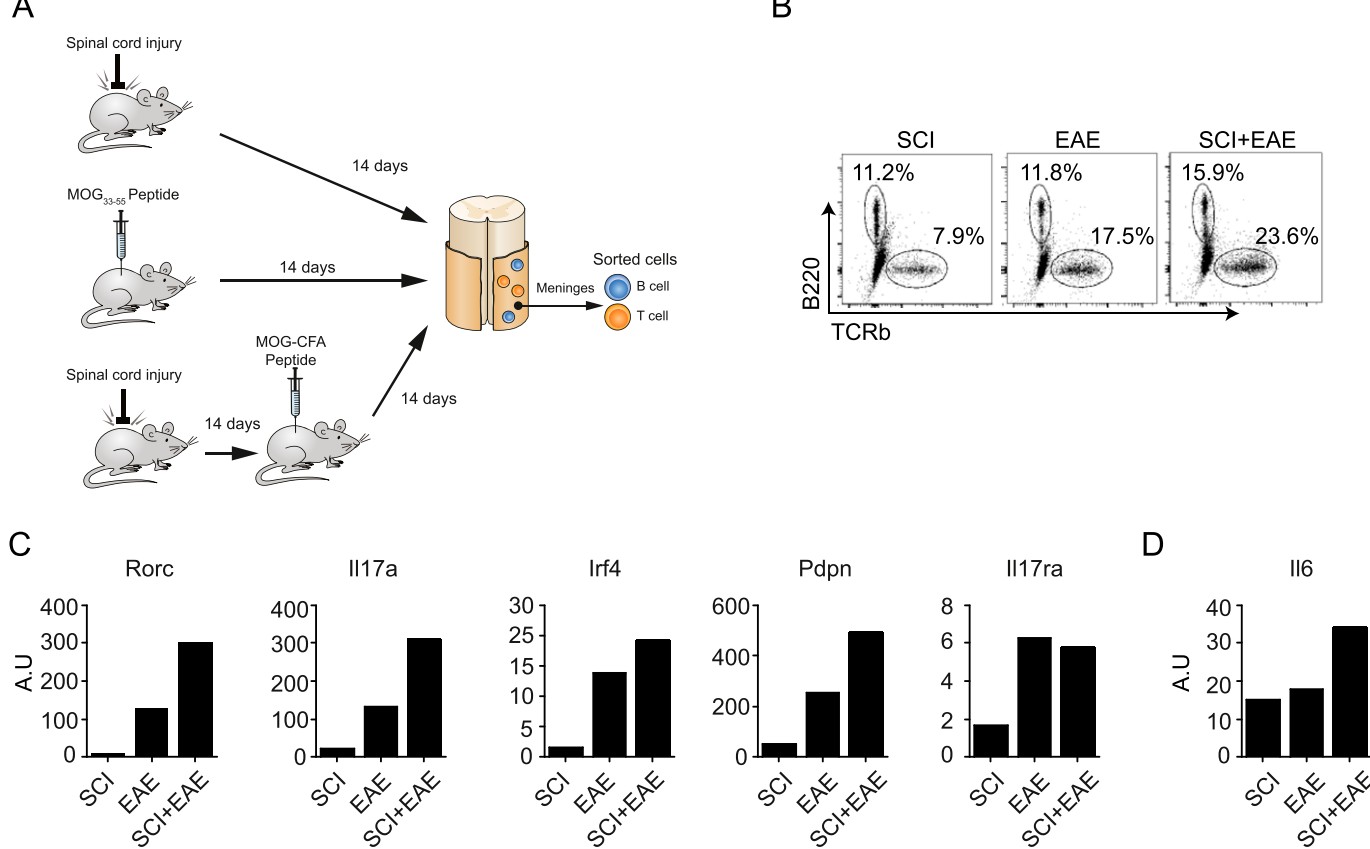

**Figure 5. Meningeal TLS following spinal cord injury affects the spine meminges inflammatory response occurring in EAE pathology.**
**(A)** A scheme depicting the experimental design of Myelin Oligodendrocyte Glycoprotein (MOG) immunization after SCI. Naïve WT mice (EAE), or mice 14 d after SCI (SCI+EAE) were immunized with the encephalitogenic MOG peptide. Animals subjected to SCI-only served as controls (SCI); cells were sorted by fluorescence activated cell sorting 14 d after MOG immunization, or after SCI (controls). **(B)** Flow cytometry plots showing TCRβ⁺ T cells and B220⁺ B cells isolated from spinal cord meninges under the different conditions detailed in (A). **(C, D)** qRT-PCR results for selected genes expressed by meningeal CD4⁺ T cells (C), and B220⁺ B cells (D) derived from spinal cord meninges under the different conditions detailed in (A). Each condition represents a pool of 10 mice.

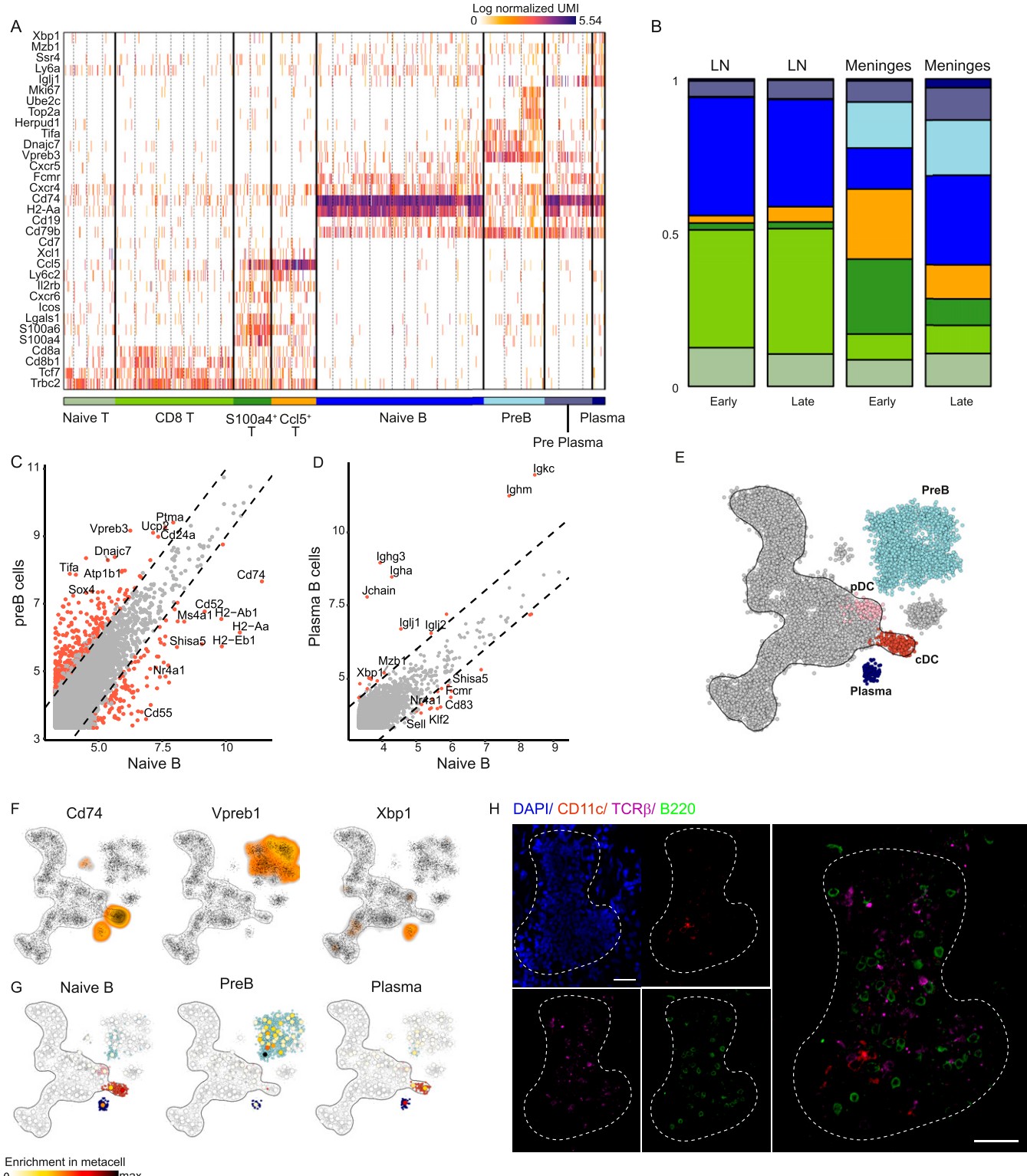

**Figure 6. Unique lymphocyte niche develops at the meninges of SOD1 mice but not at the peripheral cLN.**
**(A)** TCRβ⁺ T cells and B220⁺ B cells were single cell sorted from spinal cord meninges and cLN of SOD1 mice at different ages (92, 106, 136, and 161 d). Gene expression profiles of CD4⁺ T cells and B220⁺ B cells were annotated based on specific cellular subtypes. **(B)** Cell type distribution of CD4⁺ T and B220⁺ B cells in the spinal cord meninges (right bars) and cLN (left bars), at relatively early (92 and 106 d old) and late (136 and 161 d old) stages of disease progression. Color scale is according to cell annotation in (A). **(C, D)** Differential gene expression between naïve B cells and pre-B cells (C), and between naïve B cells and plasma B cells (D). **(E, F)** A two-dimensional representation of the metacell analysis of CD45⁺ immune cells isolated from adult mouse bone marrow (E). MetaCells related to B cell lineages were annotated according

high expression of *Ccl5^hi*, *Xcl1*, and *Il2rb*), S100a4⁺ T cells (*S100a4*, *S100a6*, and *Lgals1*), CD8⁺ T cells (*Cd8a* and *Cd8b1*), and naïve T cells (*Tcrb* and *Tcf7*). In the B-cell compartment, we identified four main cell subsets, including naïve B cells (*Cd79b*, *Cd74*, and *H2-Aa*), plasma B cells (*Xbp1*, *Mzb1*, *Iglj1*, and *Ssr4*), pre-plasma B cells, which express the *Fcmr* gene, and also exhibited a partial shared gene expression profile with plasma B cells (*Iglj1* and *Cxcr4*), and pre-B cells, which were characterized by low expression levels of genes involved in antigen presentation (*Cd74* and *H2-Aa*), but high expression of *Vpreb3* and *Dnajc7* (Fig 6A). Interestingly, we also found a cycling subset of pre-B cells with high expression of proliferative genes such as *Mki67*, *Top2a* and *Ube2c* (Fig 6A). For dynamic analysis of cell types across different time points of disease progression, we determined metacell composition in cLN compared with meninges in mSOD1 mice at early (pool of days 92 and 106) and late (pool of days 136 and 161) stages of the disease. While at the early stages of the disease, we found an enrichment of T cells relative to B cells, during the late phase of disease progression, the opposite trend appeared, and B cells accumulated in the meninges (Fig 6B). Such B-cell accumulation and dynamic lymphocyte composition was absent in the cLN (Fig 6B). A thorough cell-state analysis showed that metacell composition varied widely between tissues and across disease stages. Similar to our observations following SCI, the most prominent cellular dynamics along disease progression in SOD1 mice were observed in the meninges, rather than in the CNS-draining cLN (Fig 6B). Whereas naïve and CD8⁺ T cells were enriched in the cLN, we found massive accumulation of unique Ccl5⁺ and S100a4⁺ T-cell populations in the meninges, mainly during the early stages of disease. In the B-cell compartment, the population of pre-B cells was evident in the meninges, from early stages of disease progression onward, whereas this populaton was absent in the cLN at all tested time points along disease progression. Interestingly, whereas the pre-plasma state was present in both cLN and the meningeal compartments along all stages of disease progression, the plasma B-cell state appeared only in the meningeal compartment, and only during the late stage of the disease (Fig 6B).

Comparison of differential gene expression between naïve and activated states of the lymphocytes in the meninges showed that the meningeal lymphocyte niche is unique, enabling a specific B cell trajectory, with activation of developmental and mitochondrial genes by pre-B cells (*Vpreb3*, *Sox4*, *CD24a*, *Dnajc7*, *Tifa* and *Ucp2*; Fig 6C), and unique plasma cell and isotype switching gene programs (*Xbp1*, *Mzb1*, *Jchain*, *Ighg3*, *Ighm*, *Igkc*; Fig 6D). Differential gene expression analysis between the unique meningeal T-cell populations and the naïve T-cell state emphasized the distinct gene signature of the meningeal T-cell populations, including Ccl5⁺ cells, which highly expressed the activation markers *Xcl1*, *Nr4a2*, and *Il2rb* (Fig S2C) and S100a4⁺cells, characterized by the expression of *Lgals1*, *S100a4*, *S100a6*, and *Gzma* (Fig S2D).

The exclusive B cell niche induced in the spinal cord meninges along disease progression of ALS, in the form of pre-B cells, and the appearance of plasma B cells specifically during late stages of the disease, prompted us to thoroughly investigate and validate these meningeal novel B-cell states. To this end, we used published single-cell RNA-seq data of the mouse bone marrow hematopoietic niche (Giladi et al, 2018), and specifically, focused on the B-cell compartment (Fig 6E). By using the clustering of leukocytes isolated from mouse bone marrow along the hematopoietic process, we annotated pre-B cells (highly expressing the *Vpreb1* and *Vpreb3* genes), cDCs (expressing *Cd74*), pDCs (highly expressing *Siglech*), and plasma cells (highly expressing *Xbp1*) (Figs 6F and S2E). Interestingly, mutual nearest neighbor projection of meningeal single-cell RNA-seq data on the clustering and annotation of bone marrow hematopoiesis, revealed the stronest overlap between the pre-B and plasma cell states of B cells identified in the spine meninges during ALS progression, and in the bone marrow hematopoietic niche (Figs 6G and S2F and G). Importantly, immunostaining of mSOD1 meninges for B, T, and DC populations revealed that, as we found in spine meninges after SCI, cells tend to aggregate, and to form dense lymphocyte structures (Fig 6H).

These results emphasize that in a model of chronic spinal cord degeneration, an activated lymphocyte niche is induced within the spinal cord meninges along disease progression, representing an immune response distinct from that present in the peripheral CNS-draining cLN. The presence of meningeal lymphocytes with a unique immunological phenotype at early stages of disease progression is intriguing, but additional studies are required to determine their role, if any, in ALS etiology.

## Discussion

In the present study, we identified a unique lymphoid cell niche within the spinal cord meninges formed under several spinal cord pathologies, including SCI, inflammatory autoimmune disease (EAE), and chronic neurodegeneration (ALS).

A closer look into the meningeal composition under acute and chronic pathologies of the spinal cord revealed the formation of active meningeal lymphocyte niches with lymph node-like properties, including the presence of chemokines, blood and lymph vessels, lymphocyte segregation, and plasma cells. We view these structures as reminiscent of TLS within the meninges.

We found these meningeal TLS-like ectopic structures during the late phase of response to acute injury, when apparently moderate levels of unresolved inflammation persist within the spinal cord parenchyma (Shechter et al, 2009; Cohen et al, 2014, 2017; Raposo et al, 2014). These meningeal structures were observed in close proximity to the site of insult within the parenchyma, harboring cells with immunological properties that differed from the lymphocytes found in the spinal cord parenchyma or in the cLN. The

---

to hallmark gene expression (F). **(G)** Mutual nearest neighbor projection of meningeal naïve B, pre-B, and plasma B cells on the two-dimensional representation of the metacell analysis of CD45⁺ immune cells isolated from adult mouse bone marrow. **(H)** Representative immunofluorescent whole-mount staining of meninges derived from 145 d old SOD1 mouse. Scale bar; 30 $\mu$m. N$_{total}$ = 16 mice; n = 4–12 mice for each time point.

meningeal structures shared features reminiscent of TLS found in chronic inflammatory pathologies, including the synovium in rheumatoid arthritis, and salivary glands in Sjögren syndrome (Schroder et al, 1996; Moyron-Quiroz et al, 2004; Dieu-Nosjean et al, 2014; Pitzalis et al, 2014).

The occurrence of ectopic GC within the CNS has been mainly documented under pathological inflammatory conditions, such as those occurring in the autoimmune disease, multiple sclerosis (Magliozzi et al, 2004, 2007; Serafini et al, 2004; Peters et al, 2011). Moreover, the formation of organized lymphoid aggregates by CNS-infiltrating cells in the subarachnoid space of the spinal cord was observed after transfer of MOG-specific Th17 cells (a model of EAE) (Peters et al, 2011). Here, we found that the meningeal compartment maintains a lymphogenesis infrastructure signaling during homeostasis, and a "sterile" mechanical injury can also induce the formation of TLS in the spinal cord meninges. The meningeal lymphocyte clusters manifested several hallmarks of LN (Aloisi & Pujol-Borrell, 2006), including expression of chemokines associated with neo-lymphogenesis (Luther et al, 2000; Shi et al, 2001; Xu et al, 2003). Notably, TLS are known as ectopic lymphoid structures and represent accumulations of lymphocytes and stromal cells in an organized structure occurring outside of the SLO (Shipman et al, 2017); moreover, lymphatic vessels are pivotal players in the development of TLS and can enable drainage to the SLO.

Within the CNS, meningeal lymphatics were thoroughly studied in recent years in the dura sinuses of the brain in homeostasis (Weller et al, 2009; Aspelund et al, 2015; Louveau et al, 2015; Absinta et al, 2017), and during neuroinflammation, aging and Alzheimer's disease (Da Mesquita et al, 2018). Using in vivo magnetic resonance imaging techniques, meningeal lymphatic vessels were also identified in both humans, and in nonhuman primates (Absinta et al, 2017). Meningeal lymphatic vessels were shown to play a critical role in the trafficking of immune cells in the brain, in particular enabling T- and B-cell egress into draining lymph nodes (Louveau et al, 2015). In addition, the lymphatics might be connected to the "glymphatic" system, associated with waste management and clearance of interstitial solutes (Aspelund et al, 2015), and may also promote clearance of brain edema. Here, we suggest these meningeal lymphatic vessels can also provide the necessary support for the formation of TLS. Of note, the distance between the lesion site and peripheral SLO, might explain the need for a local ectopic lymphoid niche. Further studies regarding the origin of the meningeal lymphocytes should be performed, by examining whether they are recruited to the meninges from the periphery or from the lesion site in the spinal cord parenchyma.

The formation of a unique lymphocyte niche within the spinal cord meninges in chronic neurodegenerative conditions, as observed here along disease progression in the ALS mouse model, even before full manifestation of functional loss, and before the appearance of immune signaling in peripheral lymphatic sites, supports the existence of a regulated immune response within the CNS territory as a local immunological event that might precede the systemic immune response. This response, induced by CNS-specific mechanisms, is apparently different from those occurring in other tissues, suggesting that the meningeal response is an "ad hoc" transition station, serving as an interim communication node between the CNS and the circulation. Further studies are needed to fully elucidate the role and the effect of these niches on the fate of the parenchyma.

# Materials and Methods

### Animals

Mouse strains used is the study: C57BL/6J were obtained from Harlan Biotech; $CD11c^{DTR}$ (B6.FVB-Tg Itgax-DTR/GFP 57Lan/J), carrying a transgene encoding a human diphtheria toxin receptor (DTR) under control of the murine $CD11c$ promoter (Jung et al, 2002) were a generous gift from Prof Steffen Jung; and mSOD1 mice were obtained from animal breeding center at Ben-Gurion University of the Negev. Adult male mice aged 8–10 wk were used, unless indicated differently, as in the mSOD1 model. All animals were handled according to the regulations formulated by the Institutional Animal Care and Use Committee (IACUC).

### Scanning electron microscopy

Animals were perfused with fixative (3% PFA [Merck], 2% glutaraldehyde [EMS] in 0.1M cacodylate buffer, 5 nM $CaCl_2$, and 1% sucrose) and the meninges adjacent to the lesion site carefully dissected. After overnight post-fixation in 3% PFA and 2% glutaraldehyde, samples were fixed with 1% osmium tetroxide (EMS) in 0.1M cacodylate buffer, 5 nM $CaCl_2$, and 1% sucrose for 1.5 h. They were then incubated with 1% tannic acid (Merck) in DDW for 5 min, followed by incubation with 1% uranyl acetate (EMS) in DDW for 30 min. Samples were dehydrated by washing with ethanol at increasing concentrations, and then dried in a critical point dryer (Bal-Tec 030; Leica). After the samples were mounted onto aluminum stubs, they were sputter-coated with gold-palladium and imaged with a Zeiss Ultra 55 FEG SEM.

### Spinal cord injury

The spinal cords of deeply anesthetized mice were exposed by laminectomy at T12, and a contusive (200 kdynes) centralized injury was performed, using the Infinite Horizon spinal cord impactor (Precision Systems), as previously described (Shechter et al, 2009). The animals were maintained on twice-daily bladder expression. Animals that were contused in a non-symmetrical manner (point difference of >2 between their two hind limbs) were excluded from the experimental analysis.

### Active immunization

In the EAE induction, mice were immunized with $MOG_{35-55}$ peptide and euthanized 14 d later. To this end, 200 μg of $MOG_{35-55}$ peptide (GL Biochem Ltd.) was emulsified in incomplete Freund's adjuvant containing 0.5 mg/ml *Mycobacterium tuberculosis* (strain H37Ra; BD Diagnostics), and 200 μl of the mixture was injected subcutaneously to the mice. Pertussis toxin was not administered as part of the EAE induction.

## BrdU administration

5-Bromo-2-deoxyuridine (BrdU; Sigma-Aldrich) was dissolved by sonication in phosphate-buffered saline and injected intra-peritoneally (75 mg kg$^{-1}$ body weight) on day 13, and on the next day, 1 h before euthanizing the animals.

## Isolation of meningeal layers

Mice subjected to SCI were killed by an overdose of anesthetic followed by perfusion with PBS via the left cardiac ventricle. Spinal cord sections were cut from individual mice, including the injury site and adjacent margins (1 cm long). Spinal cord and brain meninges and parenchyma were separated by careful meningeal dissection under a binocular microscope. For the experiments involving separation of each of the three layers, identification of dura, arachnoid, and pia mater was based on their anatomical location relative to the spinal cord parenchyma and vertebra, and architecture; the dura mater was defined as the outermost thick membrane, located closest to the bones of the vertebra; the innermost pia membrane was a fibrous thin delicate membrane attached to the glial limitans enclosing the spinal cord parenchyma; and the middle arachnoid membrane was a thin, transparent membrane with a webbed sack-like appearance. Photomicrographs of spine meninges during microdissection were taken using a Universal Professional Photography HD 18× Macro Lens (Apexel) and iPhone XR (Apple).

For whole-mount staining of the meningeal tissue, the dissected meninges were immediately fixed in 4% PFA for 24 h, and then transferred to PBS containing 0.05% sodium azide.

For flow cytometry analysis and sorting of the meningeal cells for qRT-PCR analysis, the dissected tissues were digested for 45 min with 400 U/ml Collagenase type IV, at 37°C, followed by vigorous homogenization of the tissue in the staining buffer. For flow cytometry analysis and sorting of the spinal cord parenchyma, the tissue was homogenized using a software controlled sealed homogenization system (Dispomix, Medic Tools; Miltenyi), followed by resuspension in 40% Percoll for leukocyte separation.

For flow cytometry analysis and sorting of cLN lymphocytes, tissues were mechanically homogenized, and cells were washed with ice-cold sorting buffer (PBS supplemented with 0.2 mM EDTA, pH 8, and 0.5% BSA).

For single-cell sorting and analysis, cLN were digested in IMDM (Sigma-Aldrich) media supplemented with Liberase-TL (100 μg/ml; Roche) and DNase-I (100 μg/ml; Roche), and spinal cord meninges were digested in Roswell Park Memorial Institute (Sigma-Aldrich) media supplemented with collagenase IV (40 U/ml; Roche) and DNase-I (100 μg/ml; Roche). Both tissues were incubated with frequent agitation at 37°C for 20 min. After the dissociation procedures, the cells were washed with sorting buffer, filtered through 100-, 70-, and 40-μm cell strainers, and centrifuged at 380 g, for 5 min, at 4°C.

## Immunohistochemistry

Paraffin-embedded sections were used for the staining of tissue sections, which included the spinal cord parenchyma and adjacent meninges. Fixed meningeal layers were used for the whole-mount staining. Before staining, the dissected tissues were washed and blocked (20% horse serum, 0.3% Triton X-100, and PBS) for 1 h at room temperature, with shaking. Whole-mount staining with primary (in PBS containing 2% horse serum and 0.3% Triton X-100) or secondary antibodies (in PBS) was performed for 1 h at room temperature, with shaking. Each step was followed by three washes in PBS. The tissues were applied to slides, mounted with Immu-mount (Thermo Fisher Scientific Shandom), and sealed with coverslips. The following antibodies were used: rabbit anti-GFP (1:100; MBL), hamster anti-TCRβ (1:50; BioLegend), rat anti-CD3 (1:200; Serotec), rat anti-B220 (1:50; Abcam), rat anti-Ki67 (1:50; Abcam), rat anti-BrdU (1:100; Abcam), rat anti-CD31 (PECAM-1; 1:100; BD Bioscience), rat anti-pan-endothelial cell antigen (MECA-32; 1:10), rat anti-reticular fibroblast (ER-TR7; 1:100; Abcam), anti-LYVE-1 (1:100; Abcam), goat anti-CCL21 (1:50; R&D Systems), goat anti-CXCL13 (1:50; R&D systems), anti-IgD (1:100; Abcam), anti-IgM (1:100; Abcam), biotin anti-B220 (1:100; BioLegend), and armenian hamster anti-CD11c (1:100; BioLegend). The slides were exposed to Hoechst stain (1:4,000; Invitrogen Probes) for 1 min. For microscopic analysis, a Nikon light microscope (Eclipse E800) equipped with a Nikon digital camera (DS-Ri1), or fluorescence microscope (Eclipse 80i) equipped with a Nikon digital camera (DXM1200F) were used. Longitudinal sections of the spinal cord were analyzed. Evaluation of cell number was performed manually. To avoid overestimation due to counting of partial cells that appeared within the section, special care was taken to count only cells with intact morphology and a nucleus that appeared larger than 4 μm in diameter. A cluster was defined as an aggregate of 10 or more cells. Three sections from different depths were assessed for each animal in the paraffin-fixed sections; for whole-mount analysis, 3–4 meningeal samples were used.

## Flow cytometry and sorting

Cells were suspended in ice cold sorting buffer (PBS supplemented with 0.2 mM EDTA pH 8 and 0.5% BSA) supplemented with anti-mouse CD16/32 (BD Bioscience) to block Fc receptors before labelling with fluorescent antibodies against cell surface epitopes. Samples were stained using the following antibodies: PE-conjugated B220, Alexa 700–conjugated B220, Pacific Blue–conjugated B220, APC-conjugated TCRβ, FITC-conjugated TCRβ, Pacific Blue–conjugated CD4, PE-conjugated CD3, PE-conjugated CD44, PE-conjugated CD27, purified anti-mouse IgD, PE-conjugated IgM, APC-cy7–conjugated MHC-II, FITC-conjugated CD45.2, and FITC-conjugated CD4 were purchased from BioLegend; PerCPcy5.5-conjugated CD3 was purchased from eBiosciences; APC-conjugated CD62L was purchased from BD Bioscience; FITC-conjugated PNA was purchased from Sigma-Aldrich; APC-conjugated CD138 was purchased from R&D Systems; and Alexa-647-conjugated GL7 was purchased from Abcam.

For single-cell sorting, cLN and meninges-derived lymphocytes were suspended in ice-cold sorting buffer supplemented with anti-mouse CD16/32 (BD Bioscience) to block Fc receptors before labelling with fluorescent antibodies against cell surface epitopes. Samples were stained using the following antibodies: eFluor450-conjugated TER-119 and APC-conjugated CD45 were purchased from eBioscience, PE-conjugated B220 and FITC-conjugated TCRβ were purchased from BioLegend. Cells were

stained with DAPI for evaluation of live/dead cells. For sorting of T and B cells, the samples were first gated for exclusion of dead cells (DAPI⁻) and erythrocytes (TER119⁻). Cell populations were sorted with an ARIA-III instrument (BD Biosciences) and analyzed using BD FACSDIVA software (BD Bioscience) and FlowJo software (FlowJo, LLC).

Isolated live cells were single cell sorted into 384-well cell capture plates containing 2 $\mu$l of lysis solution and barcoded poly(T) RT primers for single-cell RNA-seq (Jaitin et al, 2014). Four wells were kept empty in each 384-well plate as a no-cell control during data analysis. Immediately after sorting, each plate was spun down to ensure full immersion of cells in the lysis solution, and stored at −80°C until processing.

### Quantitative real-time PCR

RNA was extracted from meningeal samples (adjacent to the lesion site, 1 cm length) using the RNeasy Fibrous Tissue Mini Kit (QIAGEN). For cell samples sorted by flow cytometry, total RNA was extracted using the miRNeasy mini kit (Quiagen). Random hexamers (AB) were used for first-strand cDNA synthesis. The procedures were performed according to the manufacturer's instructions. The relative amounts of mRNA were calculated by using the standard curve method, and were normalized to the housekeeping gene, peptidylprolyl isomerase A (PpiA). Each RNA sample was run in triplicate. Primer sequences are listed in the Supplemental Experimental Procedures section.

### MARS-seq library preparation

Single-cell libraries were prepared as previously described (Jaitin et al, 2014; Keren-Shaul et al, 2019). In brief, mRNA from cells sorted into cell capture plates was barcoded and converted into cDNA and pooled using an automated pipeline. The pooled sample was then linearly amplified by T7 in vitro transcription, and the resulting RNA was fragmented and converted into a sequencing-ready library by tagging the samples with pool barcodes and Illumina sequences during ligation, RT, and PCR. Each pool of cells was tested for library quality, and concentration was assessed as described earlier.

### MARS-seq low-level data processing

Single-cell RNA-seq libraries (pooled at equimolar concentration) were sequenced on an Illumina NextSeq 500 at a median sequencing depth of 25,274 reads per cell. Sequences were mapped to the mouse genome (mm10), demultiplexed, and filtered as previously described (Jaitin et al, 2014) with the following adaptations. Mapping of reads was performed using HISAT (version 0.1.6); reads with multiple mapping positions were excluded. Reads were associated with genes if they were mapped to an exon, using the UCSC genome browser for reference. We estimated the level of spurious unit molecular identifiers (UMIs) in the data using statistics generated from empty MARS-seq wells. Cells with less than 500 UMI, more than 500,000 UMI, or with more than 50% mitochondrial genes were excluded from analysis.

The MetaCell pipeline (Baran et al, 2019) was used to derive informative genes and compute cell-to-cell similarity, to compute K-nn graph covers and derive distribution of RNA in cohesive groups of cells (or metacells), and to derive strongly separated clusters using bootstrap analysis and computation of graph covers on resampled data. Default parameters were used unless otherwise stated. A metacell cover was produced on a combined dataset of cells from all tissues (LN and meninges) and disease states (early and late). Two-dimensional visualization of the metacell structure was performed as previously described (Baran et al, 2019). In short, a symmetric graph was constructed over all metacells, by thresholding over the co-clustering statistics (indicating how likely cells from two distinct metacells will be clustered together). This resulted in a graph with maximum degree D and any number of connected components. MetaCell computes coordinates for each metacell by applying a standard force-directed layout algorithm to the graph. It then positions cells by averaging the metacell coordinates of their neighbor cells in the K-nn graph, but filters neighbors that define a metacell pair that is not connected in the graph. MetaCell approximates the gene expression intensity within each metacell by a regularized geometric mean. It then quantifies relative expression as the log fold enrichment over the median metacell value. To annotate metacells and assign them into monocyte and macrophage states, a supervised approach was implemented, where metacells are assigned (or colored) into functional groups by expression of a curated list of marker genes. Each marker was assigned a threshold value, and all metacells whose lfp value for that marker were above the threshold were colored for that marker. In case of a conflict, a priority parameter was used to decide which marker is dominant over assignment by other markers.

Bone marrow scRNA-seq data were obtained from our previous publication (Giladi et al, 2018), describing a metacell model of 12,051 cKit⁺lineage⁻ bone marrow hematopoietic stem and progenitor cells. To project meningeal B subsets on the bone marrow scRNA-seq dataset, mutual K nearest neighbors analysis was performed, with K = 100. Neighbors were determined based on Pearson correlation over the log-transformed size-normalized UMI tables of the marker genes used to derive bone marrow clustering model. Fig 6G depicts, for each meningeal subset, all its mutual nearest neighbors in the bone marrow dataset.

### Statistical analysis

Data were analyzed using the $t$ test to compare between two groups. One-way ANOVA was used to compare several groups; the Tukey's honestly significant difference procedure was used for follow-up pairwise comparison of groups. The specific tests used to analyze each experiment are indicated in the figure legends. The results are presented as mean ± SEM.

# Data Availability

The accession number for the raw and processed scRNA-seq data reported in this article is Gene Expression Omnibus: GSE160193 Software and custom code will be available by request.

# Supplementary Information

# Acknowledgements

Research in the M Schwartz lab is supported by Advanced European Research Council grants (232835), and by the European Seventh Framework Program HEALTH-2011 (279017); Israel Science Foundation (ISF)-research grant no. 991/16; and ISF-Legacy Heritage Bio-medical Science Partnership-research grant no. 1354/15. The research of I Amit is supported by the Seed Networks for the Human Cell Atlas of the Chan Zuckerberg Initiative and by Merck KGaA, Darmstadt. I Amit is an incumbent of the Eden and Steven Romick Professorial Chair, supported by the Howard Hughes Medical Institute International Scholar Award, the European Research Council Consolidator Grant (no. 724471-HemTree2.0), an melanoma research alliance Established Investigator Award (no. 509044), Deutsche Forschungsgemeinschaft (DFG) (no. SFB/TRR167), the Ernest and Bonnie Beutler Research Program for Excellence in Genomic Medicine, the Helen and Martin Kimmel awards for innovative investigation, and the award of the Wolfson Foundation and Family Charitable Trust. The Thompson Family Foundation Alzheimer's Research Fund and the Adelis Foundation also provided support. RG Lichtenstein is supported by Keren Hayesod, Magbit France.

## Author Contributions

M Cohen: conceptualization, data curation, formal analysis, supervision, validation, investigation, visualization, methodology, project administration, and writing—original draft, review, and editing.
A Giladi: data curation, software, validation, and investigation.
C Raposo: conceptualization, formal analysis, validation, investigation, methodology, and writing—original draft, review, and editing.
M Zada: investigation.
B Li: investigation.
J Ruckh: investigation.
A Deczkowska: investigation.
B Mohar: investigation.
R Shechter: conceptualization.
RG Lichtenstein: conceptualization, resources, supervision, and investigation.
I Amit: conceptualization, resources, supervision, and investigation.
M Schwartz: conceptualization, resources, data curation, formal analysis, supervision, funding acquisition, validation, investigation, visualization, methodology, project administration, and writing—original draft, review, and editing.

## Conflict of Interest Statement

The authors declare that they have no conflict of interest.

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
