## [Reviewer comments · Life Science Alliance]

Life Science Alliance

Meningeal lymphoid structures are activated under acute and chronic spinal cord pathologies

Merav Cohen, Amir Giladi, Catarina Raposo, Mor Zada, Baoguo Li, Julia Ruckh, Aleksandra Deczkowska, Boaz Mohar, Ravid Shechter, Rachel G Lichtenstein, Ido Amit and Michal Schwartz

DOI: <https://doi.org/10.26508/lsa.202000907>

Corresponding author(s): Michal Schwartz and Merav Cohen, Weizmann Institute of Science, Israel

Review Timeline:

Submission Date:	2020-09-15
Editorial Decision:	2020-09-15
Revision Received:	2020-10-04
Editorial Decision:	2020-10-06
Revision Received:	2020-10-27
Accepted:	2020-10-28

Scientific Editor: Shachi Bhatt

Transaction Report:

Please note that the manuscript was previously reviewed at another journal and the reports were taken into account in the decision-making process at Life Science Alliance. Since the original reviews are not subject to Life Science Alliance's transparent review process policy, the reports and author response cannot be published.

Re: Life Science Alliance manuscript #LSA-2020-00907-T

Prof. Michal Schwartz
Weizmann Institute of Science
Neurobiology
234 Herzl St.
Rehovot 7610001
Israel

Dear Dr. Schwartz,

Thank you for submitting your manuscript entitled "Meningeal tertiary lymphoid structures are formed under acute and chronic spinal cord pathologies" to Life Science Alliance (LSA). The manuscript was assessed by expert reviewers, whose comments are appended to this letter.

For a brief overview, the manuscript was reviewed at another journal and the authors transferred the manuscript along with the reviews. The reviewers were concerned that the paper about cross-contamination during separation of the 3 meningeal layers and the the lack of sufficient experiments to qualify these as germinal centers. After assessing the manuscript, reviewers' comments, and point-by-point response from the authors, the LSA editors deemed it to be appropriate for publication, provided the authors tone down their conclusions and address the following remaining points -

- + please edit the text to clarify that the possibility of cross-contamination between the 3 leptomeningeal layers has not been completely eliminated (Rev 1 pt 1)
- + if possible, please perform additional confirmatory stainings for the lymphatics (Rev 1 pt 2). While these data would not be required for publication at LSA, we encourage you to include them in the revised manuscript. If unavailable, please tone down the respective conclusion accordingly
- + please provide clarifications for the minor concerns raised by Rev 1 in pts 3 & 4
- + please provide additional staining for GC markers (e.g. Bcl6, AID, possibly GL7 or follicular dendritic cell markers like CR2 (Rev 2) - THESE DATA WOULD BE REQUIRED FOR PUBLICATION IN LSA
- + please provide a no-BrdU control for Fig 4H (Rev 2), or an explanation for why that would be out of scope for this study
- + please tone down the conclusions given that the SCI+EAE group could not be monitored beyond 14 days (Rev 2)
- + if possible, please provide additional views for Fig 3C in support of the claim that CCL21 is expressed by ERTR7+ cells, or provide an explanation for why the image does not represent what text claims (Rev 1 pt 4 and Rev 2)
- + please adjust the data presentation to show reproducibility of measurements or error bars

We would be happy to discuss the individual revision points further with you should this be helpful. While you are revising your manuscript, please also attend to the below editorial points to help expedite the publication of your manuscript. Please direct any editorial questions to the journal office. The typical timeframe for revisions is three months. Please note that papers are generally considered through only one revision cycle. When submitting the revision, please include a letter addressing the reviewers' comments point by point.

Thank you for this interesting contribution to Life Science Alliance. We are looking forward to receiving your revised manuscript.

Sincerely,

Shachi Bhatt, Ph.D.
Executive Editor
Life Science Alliance

B. MANUSCRIPT ORGANIZATION AND FORMATTING:

RE: Life Science Alliance Manuscript #LSA-2020-00907-TR

Prof. Michal Schwartz
Weizmann Institute of Science
Neurobiology
234 Herzl St.
Rehovot 7610001
Israel

Dear Michal,

Thank you for submitting your revised manuscript entitled "Meningeal lymphoid structures are activated under acute and chronic spinal cord pathologies". We would be happy to publish your paper in Life Science Alliance (LSA) pending final revisions necessary to meet our formatting guidelines.

Along with the points listed below, please also address the following:

- please add ORCID ID for both corresponding authors-you should have received instructions on how to do so
- please add a conflict of interest statement to your main manuscript text
- please use the [10 author names, et al.] format in your references (i.e. limit the author names to the first 10)
- please add the supplementary figure legends to the main manuscript text (directly under the main figure legends)
- please make sure the order of the manuscript sections complies with LSA's formatting guidelines (<https://www.life-science-alliance.org/manuscript-prep#format>)
- please make sure the insets for Figures 1A, 1B and 2A match with the zoomed in images in the same panel
- please update the email contact information of two of the authors (LSA editorial team is already in touch with the corresponding author for this)
- as per LSA's guidelines, large scale data sets like MARS-seq are required to be deposited in the relevant database (<https://www.life-science-alliance.org/manuscript-prep#datadepot>), and this information and the accession number need to be included in a separate Data Availability section in the manuscript

A. FINAL FILES:

B. MANUSCRIPT ORGANIZATION AND FORMATTING:

Sincerely,

Shachi Bhatt, Ph.D.

Executive Editor

Life Science Alliance

<https://www.life-science-alliance.org/>

Tweet @SciBhatt @LSAjournal

RE: Life Science Alliance Manuscript #LSA-2020-00907-TRR

Prof. Michal Schwartz
Weizmann Institute of Science
Neurobiology
234 Herzl St.
Rehovot 7610001
Israel

Dear Dr. Schwartz,

Thank you for submitting your Research Article entitled "Meningeal lymphoid structures are activated under acute and chronic spinal cord pathologies". It is a pleasure to let you know that your manuscript is now accepted for publication in Life Science Alliance. Congratulations on this interesting work.

*** The manuscript is still missing the GEO accession number for the RNA-Seq data. Please make sure to add that at the proofs stage. We have notified our copyeditors of the same.***

As per your request, Reviews and point-by-point responses associated with peer-review of this manuscript will not be published alongside the manuscript. If you do not want to opt out of having the reviewer reports and your point-by-point responses displayed, please let us know immediately. Only editor's decision letters will be published.

DISTRIBUTION OF MATERIALS:

Again, congratulations on a very nice paper. I hope you found the review process to be constructive and are pleased with how the manuscript was handled editorially. We look forward to future exciting submissions from your lab.

Sincerely,

Shachi Bhatt, Ph.D.

Executive Editor

Life Science Alliance

<https://www.lsajournal.org/>

Tweet @SciBhatt @LSAjournal